# The Potential Utilization of High-Fiber Agricultural By-Products as Monogastric Animal Feed and Feed Additives: A Review

**DOI:** 10.3390/ani11072098

**Published:** 2021-07-15

**Authors:** Wen-Yang Chuang, Li-Jen Lin, Hsin-Der Shih, Yih-Min Shy, Shang-Chang Chang, Tzu-Tai Lee

**Affiliations:** 1Department of Animal Science, National Chung Hsing University, Taichung 402, Taiwan; xssaazxssaaz@yahoo.com.tw; 2School of Chinese Medicine, College of Chinese Medicine, China Medical University, Taichung 404, Taiwan; linlijen@mail.cmu.edu.tw; 3Taiwan Agricultural Research Institute, Council of Agriculture, Executive Yuan, Taichung 413, Taiwan; tedshih@tari.gov.tw; 4Hsinchu Branch, Livestock Research Institute, Council of Agriculture, Miaoli, Hsinchu 368, Taiwan; emshy@tlri.gov.tw; 5Kaohsiung Animal Propagation Station, Livestock Research Institute, Council of Agriculture, Pîntong 912, Taiwan; macawh@tlri.gov.tw; 6The iEGG and Animal Biotechnology Center, National Chung Hsing University, Taichung 402, Taiwan

**Keywords:** agriculture by-product, feed additive, animal health, fermentation

## Abstract

**Simple Summary:**

High-fiber agriculture by-products, which can enhance animal performance and health, have the potential to be used as feed additives. Before using high-fiber agriculture by-products, it is necessary to pay attention to the problem of anti-nutritional factors and contamination due to mycotoxins. Solubility and fermentability are the keys that mainly affect fiber availability. In recent years it has been pointed out that fiber as an animal feed or feed additive does not seem to be as unfeasible as previously thought. Instead, dietary fiber and other functional compounds, such as polyphenol and flavonoids, could enhance health, antioxidant capacities, and stabilize the microbiota in animals. In addition, high-fiber agriculture by-products are a suitable and inexpensive source of fiber and their proper use may reduce costs of animal feeding. Scientists must integrate characteristics and appropriate usage analysis to jointly evaluate the effects of different fiber compositions on those animals. Based on this foundation, animal producers should be encouraged to use high-fiber agricultural by-products as animal feed and feed additives.

**Abstract:**

With the increase in world food demand, the output of agricultural by-products has also increased. Agricultural by-products not only contain more than 50% dietary fiber but are also rich in functional metabolites such as polyphenol (including flavonoids), that can promote animal health. The utilization of dietary fibers is closely related to their types and characteristics. Contrary to the traditional cognition that dietary fiber reduces animal growth, it can promote animal growth and maintain intestinal health, and even improve meat quality when added in moderate amounts. In addition, pre-fermenting fiber with probiotics or enzymes in a controlled environment can increase dietary fiber availability. Although the use of fiber has a positive effect on animal health, it is still necessary to pay attention to mycotoxin contamination. In summary, this report collates the fiber characteristics of agricultural by-products and their effects on animal health and evaluates the utilization value of agricultural by-products.

## 1. Introduction

The world’s demand for food is increasing, and this includes bulk cereals including wheat, rice, corn, and soybean [1]. However, crop production is accompanied by large amounts of agricultural by-products. The by-products produced during agricultural production mainly include the stems, hulls, leaves, brans, and roots of plants. In the past, these agricultural by-products were mostly incinerated or composted to be used as fertilizer, but they also caused the problem of air pollution and inefficient use of resources [2]. The demand for animal protein is also rising, with a concomitant increase in the demand for animal feed. This further increases the requirement for common feed crops and their by-products. However, with increasing awareness of environmental protection, agricultural waste needs to be properly treated [3]. Agricultural by-products from different plant sources may also contain dietary fiber (such as cellulose and hemicellulose), starch, crude protein, oligosaccharides or vitamins, and other nutrients [1,3]. Phenols and flavonoids, the most important phytochemicals, are often included in the fiber [3,4].

Carbohydrates, one of the main sources of energy for animals, can be roughly divided into two broad categories. The first category is the source of energy for animals, such as starch, glucose, and sucrose, which are decomposed by enzymes produced by animals. The second category is the dietary fibers cellulose, chitin, and other fibers which may be fermented by the microorganism. In the past, scientists and animal producers considered that the role of dietary fiber in feed is mainly to dilute protein or energy [5]. However, studies in recent years have often pointed out the complex role of fiber in animal feed. High-fiber and low-fat diets can reduce the incidence of cardiovascular disease and promote intestinal health [6]. In addition, dietary fiber content positively correlates with microorganisms such as *Bifidobacterium*, *Roseburia*, and *Eubacterium rectale*, and these microorganisms are the primary microorganisms secreting short-chain fatty acids (SCFA) which could provide energy for intestinal epithelial cells [7,8]. When there is a single source of carbohydrates, it can easily cause a decrease in the diversity and richness of the intestinal microbiota [9]. It even affects mucosal secretion and the health of intestinal epithelial cells.

Fiber, like *Pennisetum* grass, a kind of common forage species, can also be used as a medium for mushroom cultivation [3,10]. Previous studies showed those mushroom wastes compost (high-fiber agricultural by-products) can reduce the inflammatory response and increase the antioxidant capacity of serum and liver of animals and enhance the growth of intestinal villi when added in the animal’s diet [3,6,10]. Moreover, adding dietary fiber (as potato pulp, sugar beet pulp, and pectin residue) can also promote the gastric emptying rate and alter the satiety of animals, the author’s results can also help sows maintain the shape of feces and reduce the incidence of diarrhea by water-retaining contents [11]. However, the sources and types of fibers also greatly limit the availability of fibers [12]. Fibers with too high a molecular weight or degree of polymerization may be difficult to digest and be utilized by the animal’s gut microbiota in a short time [13]. Too many soluble non-starch polysaccharides (NSP) may lead to excessive fermentation by intestinal microorganisms and reduce animal production performance [14]. Likewise, improperly preserved fibers may also become a breeding ground for the growth of fungi and cause the accumulation of mycotoxins. Fortunately, fiber can be fermented under controlled conditions through probiotics or enzymes, thereby reducing the damage caused by anti-nutritional factors or toxins in the fiber to animals [15,16,17].

Overall, although fiber is generally beneficial for animal health, one must pay attention to the characteristics of the fiber, including solubility, fermentability, or anti-nutritional factors, as these affect the timing or availability of fiber. In vitro, fermentation of fibers with probiotics (Saccharomyces cerevisiae and Aspergillus oryzae) and/or enzymes (phytase) can form postbiotics with fermentation products and increase fiber availability [17]. This article aims to discuss the availability and possible harm of high-fiber agricultural by-products in animal feed.

## 2. Potential Utilization of Feed Crop By-Products

In the feed industry of animal husbandry, corn, soybean, and wheat are the main feed sources. To produce these feed ingredients, tons of by-products, such as straw, hull, and bran, also need to be produced. Corn grains only account for about 20% [18] and soybeans account for about 30% of the dry whole plants. Accordingly, the residues are burnt or used as a material for compost, not only exacerbating air pollution but also causing waste of resources [10,15]. A wastage of these residues is not consistent with the concept of sustainable development. On the other hand, the concept of animal welfare is rising, and therefore, the producers have to try to enhance the health and decrease stress when rearing animals. Fiber increases sow satiety, decreases aggressive behavior, and improves antioxidant capacities [10,11], fulfilling the requirements of animal welfare. Therefore, feed crop by-products could be a high potential feed ingredient of the animal feed industry.

## 3. Composition of Fiber

Fiber from food sources is closely related to animal health; depending on the source, fiber can be divided into composite dietary fiber or a purified prebiotic. According to Prasad and Bondt [12], dietary fiber is defined as “non-digestible polysaccharides largely composed of complex carbohydrates”. The International Scientific Association for Probiotics and Prebiotics (ISAPP) defines a prebiotic as “a substrate that is selectively utilized by host microorganisms conferring a health benefit” [19]. In the field of animal nutrition, scientists classify fibers based on their biochemical characteristics into acid detergent fiber (cellulose and lignin), neutral detergent fiber (hemicellulose, cellulose, and lignin), and dietary fiber (non-starch polysaccharides) (Figure 1).

Nevertheless, fermentability and solubility are also some of the concerns of animal nutritionists. The fermentability of fiber is related to the microbial composition of the intestine and short-chain fatty acids, while the solubility is related to the satiety and fecal configuration of animals [11,12]. Insoluble fibers generally include cellulose and lignin, while soluble fibers include inulin and pectin [12]. Interestingly, most soluble fibers are more fermentable than insoluble fibers. Therefore, soluble fiber (3% pectin) may also promote excessive growth of microorganisms (total count of bacteria and *E. coli*) and increase chyme viscosity [14].

Considering cost constraints, it is more difficult to use highly processed and purified prebiotic as feed additives for animal production; on the contrary, dietary fiber is very suitable. However, dietary fiber (consisting of approximately 40–50% cellulose, 20–30% hemicellulose, 10–25% lignin, and about 35% pectin) may contain dozens or even hundreds of different carbohydrate bonding patterns, as well as complex compositions such as phenolic, minerals, and vitamins [20]. Therefore, most of the research on fiber is still mainly aimed at prebiotics. Research on dietary fiber indicates that it is difficult to explain which kinds of fibers have positive or adverse effects, and only a simple distinction can be made according to the source of dietary fiber. Although there is almost no relevant research reported, to understand the efficacy of different fibers in complex dietary fiber, metabolomic techniques could assist scientists in understanding the changes in animal digestive state after involving dietary fiber [7,21].

## 4. Partial Plant Phytochemicals and Anti-Nutrition Factors in High-Fiber By-Products

In addition to fiber, plants also contain a variety of secondary metabolites, including antioxidant peptides, phenols, flavonoids, phytic acid, trypsin inhibitors, and lectins [22,23]. These plant-derived secondary metabolites are rich in antioxidant, anti-inflammatory, and antibacterial activity, but may also reduce the animal’s absorption efficiency to nutrients such as minerals or amino acids [22]. Phytochemicals have been shown to have positive antioxidant benefits toward animals in terms of favored growth performance, production quality, and enhanced endogenous antioxidant systems, possibly by directly affecting specific molecular targets or/and indirectly as stabilized conjugates affecting the metabolic pathways [11,13,23]. Accordingly, dissecting the antioxidant effects and the underlying mechanism of dietary phytochemicals is an important area. Therefore, much attention is being focused on a new wave of nutrigenomics [6,10,23]. Several studies have been dedicated to understanding and formulating mechanistic pathways by which these naturally derived substances could alter the fate of cells, particularly the antioxidant properties of phytochemicals have been implicated as stress-alleviation agents [11,23,24]. Besides phytic acid, trypsin inhibitors, and lectins, there are also anti-nutritional factors commonly found in plants [24,25] that cannot be ignored while using high-fiber agricultural by-products as animal feed or feed additive.

### 4.1. Phenol

Phenols are common components in plants, and their basic unit structure is a benzene ring connected with an OH-. In general, the phenolic compounds found in plants vary from plant to plant. Some of the well-known phenols include catechins, chlorogenic acid, caffeic acid, and quinine [26]. These phenolic compounds improve the antioxidant capacity of animal serum and reduce the degree of animal inflammation [27]. On the other hand, some phenolic compounds can also inhibit the production of pathogenic bacteria such as grape extract, which was effective at inhibiting antibiotic-resistant *Staphylococcus aureus* and *E coli*, including methicillin-resistant *S. aureus*, with minimum inhibitory concentrations (MIC) ranging from 0.3 to 3.0 mg/mL, by chelating minerals necessary for the microorganism survival, or perforating the microbial cell membranes [28]. However, some plants also contain potentially toxic phenolic compounds, such as Pyrogallol, a simple phenolic found in green tea, which was shown to cause hepatic damage when administered at 100 mg/kg in rats. Serum enzymes including aspartate aminotransferase and alanine aminotransferase (ALT) as well as malondialdehyde (MDA) were increased, suggesting that free radical formation and pro-oxidant toxicity played a role. Therefore, excessive supplementation may cause animal poisoning, shock, and reduced nutrient absorption in animals [29].

### 4.2. Flavonoids

Flavonoids are a more functional classification of phenolic compounds, and their basic unit structure is 2-phenyl-1,4-benzopyrone. The classic flavonoids include estrogen, anthocyanin, and catechins [30]. The function of flavonoids is similar to that of phenols. Among them, catechin, the classic flavonoid in green tea, can increase the antioxidant capacities, and also promote fat metabolism in animals, thereby reducing low-grade inflammation caused by excessive accumulation of fat [30,31]. Catechins can be found not only in tea but also in *Pennisetum* [3]. Phenols or flavonoids, the plant secondary metabolites (functional phytochemicals) which have the ability to induce the expression of antioxidant/phase II enzymes, appear to have a major role in acting as modifiers of signal transduction pathways to elicit its cytoprotective responses through suppressing stress-induced protein activation and enhancing Kelch-like ECH associating protein 1 (Keap1), a cytoskeleton binding protein, and dissociation from nuclear factor (erythroid-derived 2)-like 2 (Nrf2) in response to stressors. Therefore, suppression of abnormally amplified oxidation signaling and restoration of improperly working systems, as well as the activation of antioxidant machinery could provide important strategies for prevention of oxidative stress and augmentation of antioxidant defense in animals [3,15].

### 4.3. Gossypol

Gossypol is a type of di-sesquiterpene aldehyde, mainly found in cotton seeds [32,33]. Besides having high antioxidant and antitumor activity, gossypol can reduce the inflammatory response by decreasing the NF-κB expression, and also has high neurotoxicity and reproductive toxicity [33,34,35]. Therefore, the residual amount and use of gossypol in plant raw materials are highly valued. Owing to its drought-tolerant properties, cotton is particularly suitable for planting in arid areas [36] leading to the production of cottonseed oil and cottonseed meal. However, as an agricultural by-product, cottonseed meal contains gossypol in addition to its high fiber content. Using cottonseed meal, such as Hy-line Brown layer or Shanshui White duck, will leave gossypol in the eggs, reduce the yolk quality, and destroy duck liver cells [37,38]. In contrast, research on a ruminant (*Odocoileus virginianus*) indicated the accumulation of gossypol in the white-tailed deer serum when using whole cottonseed supplements but did not affect white-tailed deer health and reproduction [39]. Overall, the utilization of cottonseed meals should be strictly controlled and the hazards of cottonseed phenol residues on animal health and food quality must be closely monitored.

### 4.4. Tannic Acid

Tannic acid is a water-soluble polyphenolic substance and is found naturally in plants. Tannic acid has antioxidant, anti-inflammatory, and antibacterial activities, but may also reduce the digestive enzyme activity of animals, hinder mineral absorption, and exhibit cell membrane toxicity [40,41]. A higher tannic acid addition also reduced the feed intake of broilers [42,43].

Using drinking water to ingest 10 g of tannic acid in poultry greatly reduces the utilization of amino acids, especially methionine, histidine, and lysine, and D-xylose [43,44]. Tannic acid also reduces plasma iron levels and the performance of weaned piglets [45]. Mansoori and Modirsanei [46] further pointed out that tannic acid reduces the effect of the anti-coccidial vaccine. In contrast, Tonda et al. [47] indicated that the addition of 0.5 g/kg gallnut tannic acid could decrease the number of *Eimeria* spp. Overall, tannin is regarded as a plant-derived anti-nutritional factor. However, because Kubena et al. [42] reported the antibacterial activity of tannic acid, Cengiz et al. [48] added some (2 g/kg feed) tannic acid to animal feed and found that it could reduce the negative effects of barley NSP. Therefore, if the appropriate amount of tannic acid is found through animal trial for evaluation, it will have a positive effect on animal health.

### 4.5. Phytic Acids

About 50–80% of plant phosphorus is stored in phytic acid, mainly in the form of inositol hexakisphosphate (IP6) [49]. There are different types of phosphoric acid depending on the number of phosphates attached to inositol [49,50]. Phosphoric acid on inositol can combine with minerals or amino acids of valence 2 or 3 and reduce the absorption rate of nutrients and enzyme activity of animals [49,50,51,52]. In addition, the involvement of too much phytic acid may also lead to poor phosphorus utilization by animals, causing the accumulation of phosphorus in stool and this may cause pollution to the environment [53]. Fortunately, phytic acid can be degraded by phytase. Currently, most commercial phytase is produced from fungi or *Escherichia coli* [50]. Adding phytase to animal feed can reduce the impact of phytic acid on animal digestibility [54].

### 4.6. Trypsin Inhibitor

Trypsin inhibitor is a well-known anti-nutrition factor in plant-based ingredients, which could reduce the degradation of protein and thereby reduce the nutrient absorption of animals [55]. Fortunately, trypsin inhibitors can be reduced or destroyed by heat and reducing agents [55]. Therefore, the process methods significantly affect the quality of the feed source containing a high amount of trypsin inhibitor, such as that of soybean, chickpea, and buckwheat [56]. Avilés-Gaxiola et al. [57] reported that while reducing agents, such as L-cysteine, can decrease the activities of trypsin inhibitor (up to 89.1%), treating the raw ingredients by heat and reducing agents could be more effective (up to 99.4%). Another study showed that polyphenols extracted from tea, especially *epigallocatechin gallate* and *epigallocatechin*, can suppress the activities of trypsin inhibitors [58].

### 4.7. Lectin

Lectins are commonly produced plant proteins and can chelate the carbohydrate on the glycoprotein, thus they can condense and destroy cells [59]. Ricin, one of the most well-known lectins, is extracted from *Ricinus communis* and causes animal death [59]. Many common feed ingredients also contain lectins, including corn, soybeans, and wheat [59]. The toxicity of lectin is mainly based on the “degree of lectin resistance to proteolytic degradation” [59]. Accordingly, the toxicity of lectins can also be utilized for removing hazardous waste in animals. Therefore, for the utilization of lectins, their physicochemical properties (including hemagglutination activities, inflammatory, antibacterial, and antifungal activity) as well as dosage must be considered and evaluated [60].

## 5. Effect of Fiber Addition on Animal Production Performance and Microbiota

Animals cannot effectively degrade dietary fiber, so the fiber ingested by the animal initially undergoes preliminary fermentation and degradation through the intestinal microbiota before being used as an animal’s nutritional source. When digesting fiber, the composition of microbiota might therefore alter [7]. For instance, excessive intake of soluble NSP may lead to the increase of *E. coli* and *Clostridium*, which negatively impact animal performance [14]. Conversely, insoluble dietary fiber may absorb harmful substances and excrete them [12]. Animals have different sensitivities in the gut microbiota at different growth stages, therefore, animals with different life cycles should be discussed separately. In this review, we mainly discuss the effect of dietary fiber on the changes of animal intestinal microbes and production performance. Table 1 lists the effect of different fiber replacement or addition on animal growth performance and microbial microbiota in the ileum.

### 5.1. Effects of Fiber on Broilers

Mateos et al. [66] indicate that the dietary fiber addition could improve poultry health, including increase the gizzard weight, nutrition digestibility, and intestine morphology; however, the fiber addition should not be over 3% in broiler diet. Kermanshahi et al. [14] indicated that the 3% cellulose addition in the diet of broilers (Ross 308) had similar growth performance, intestinal morphology, microbe composition, and serum characteristics in comparison to the control group during the experimental period (0 to 14 days). However, the 3% pectin and carboxymethyl cellulose addition would increase the number of *E. coli* and thereby decrease the growth performance and gut health.

### 5.2. Effects of Fiber on Swine

Pigs have different nutritional requirements in different stages of their life cycle. For piglets during lactation, sow milk is the most important source of energy; therefore, it is important to maintain stable production of sow milk and ensure that every piglet consumes milk [65,67]. Piglets face huge environmental changes and stress when weaning, while for the piglets that have just been weaned, there are significant changes in the type of food ingested, so diarrhea is prone to occur [45,64]. Feeding piglets after weaning provides an appropriate amount of insoluble fiber (as inert fiber) to avoid diarrhea caused by accumulation of undigested nutrients and help piglets to restore intestinal function [64,65]. Moreover, it is necessary to maintain stable feeding and improve the physiological health of pigs during their late growth period or during sow pregnancy. Therefore, adding dietary fiber to the sow diet could increase feed intake during lactation, thereby increasing the number of weaned piglets, weaned piglet weight, and average daily gain. Feeding high-fiber diets in late pregnancy may also reduce the stillbirth rate by reducing delivery time. In addition, it can alleviate constipation in pregnant and lactating sows [67,68].

The addition of 40 mg/kg chito-oligosaccharide can improve the carbohydrate composition of sow milk, and therefore increase litter size, survival rate, and total litter weight of both at 12 and 21-day-old pigs [67]. In total, simultaneous addition of 1% soluble (inulin) and insoluble (lignocellulose) fiber could significantly increase the feed conversion rate of weaning pigs (24 to 52-day-old); however, adding soluble fibers or insoluble fibers singly is less effective [64]. The total short-chain fatty acids increase by about 30% in all 1% fiber addition groups compared to the control group, thereby enhancing the tight junction (TJ) expression in piglet ileum. Nevertheless, the sodium-glucose cotransporter-1 gene expression in piglet jejunum increased by 2-fold in the 1% insoluble fiber addition group compared to that in the control group [64]. Moreover, Naya et al. [68] indicated that the addition of soluble fiber could decrease tail biting in 3 to 10-week-old pigs. According to the above-mentioned results, insoluble fiber is better than soluble fiber in promoting the growth performance of pigs. However, adding both soluble and insoluble fibers is more effective than adding insoluble fibers singly.

## 6. Effect of Functional Components of Fibers on Animal Physiology

### 6.1. Fibers Antioxidant and Anti-Inflammatory Responses in Animals

Previous studies have discussed in detail the inflammation and antioxidant mechanism of phytochemicals on animal immune regulation and antioxidant capacity [10,69,70]. Briefly, when animals are subjected to environmental stress, such as heat stress or pathogens infection, stimulation causes an inflammatory response and increases oxidative stress in animals [71]. Stimulated by pathogenic bacteria, the animal initiates an immune response leading to a cytokine storm [71]. However, excessive inflammation can reduce animal performance and even lead to death [10]. On the other hand, when the oxidative pressure is too high, animals are not able to eliminate the damage caused by free radicals to cells or organs [69]. Among them, the animal’s antioxidant system is mainly regulated by the liver, so the antioxidant capacity is also related to liver performance [3]. Applying phytochemicals or botanical compounds to the feed could promote intestinal health, reduce the inflammatory response, and enhance the antioxidant capacity of the animal [10]. The mechanisms are mainly the nuclear factor (erythroid-derived 2)-like 2 (Nrf2) and nuclear factor kappa B (NF-κB) for they are respectively the key transcription factors involved in oxidative stress and inflammation for elucidating the underlying signal transduction pathways. Therefore, phytochemicals can regulate these transcription factors leading to the improvement of oxidative status, the heme oxygenase-1 (HO-1) gene is found to be crucial for Nrf2-mediated NF-κB inhibition. Hence, proper fibers as phytochemicals (likely 0.5–1% mulberry leaves addition in laying hens) with such modulatory effects should be used to explore the possible crosstalk in oxidative stress and immunomodulation in animals [69,70,71].

### 6.2. Fibers Satiety in Animals

Obesity caused by overfeeding of sows is not conducive to piglet production, because it may increase the production time of sows, and suffocate oversized piglets in the vagina, reduce the number of births, and overall life cycle of sows [72]. Therefore, during the sow’s pregnancy, the breeder limits the feed intake of the sow [73]. However, failure to get enough satisfaction may lead to a more stereotyped and aggressive behavior of the sow [73,74].

To improve the satiety of sows, scientists have proposed that high-fiber feed can be given to dilute the total energy in the feed [74,75]. Although dietary fiber cannot be digested by endogenous enzymes in animals, gut microbes degrade dietary fiber into short-chain fatty acids, further regulating animal feeding patterns and gut health [76]. According to its characteristics, fiber can be either soluble or fermentable. After dissolving, soluble fiber increases the viscosity of chyme, thereby increasing the transit time of chyme [11]. Therefore, compared with insoluble fiber, soluble fiber can improve the satiety of animals [11]. The satiety of food in the liquid phase is higher than that in the solid phase [11]. Fermentable fiber can produce more short-chain fatty acids, stimulate animals to produce antimicrobial peptides, and further adjust intestine health [76].

Giving high-fiber (totally dietary fiber about 28.2%), low-energy diets (including mainly 24.4% soybean hulls) to sows in group cultures can improve their feeding time and health, and reduce aggressive and stereotyped behavior [73,74]. Providing higher amount of fiber (7.5% crude fiber consisting of 20% Alfalfa meal and 52% corn in the lactation diet) can also improve the welfare and improve the production performance of sows [77] and, therefore, be beneficial to the health of pregnant sows.

## 7. Fibers Can Increase Animal Performances

The physiological response of fiber addition to animals has been discussed in detail in the earlier sections. In this review, we discuss further how fiber addition and application could improve animal performances. Traditionally, fiber is considered to be an anti-nutritional factor in feed and has a negative effect on animal palatability and production performance [5]. In contrast, many studies have repeatedly pointed out that adding fiber can improve animal performance [78,79,80,81,82,83]. Herein, the review discusses more the role of fiber in increasing animal production performance by promoting intestinal health, immune regulation, and changing fat metabolism patterns. Table 2 lists the effect of different fiber supplements on digestibility, health, or production of animals.

### 7.1. Intestinal Health and Immune Regulation

Both mice and poultry studies have pointed out that high fiber intake can increase the performance and thickness of the intestinal barrier of animals, including TJ and mucosal proteins [7,16,85,86,87]. The addition of prebiotics (autoclaved drinking water supplements with 1.0% oligofructose-enriched inulin (*w*/*v*)) can also increase the viscoelasticity of mucosal proteins [88]. A robust intestinal barrier can increase the distance between pathogenic bacteria in the intestinal cavity and intestinal epithelial cells and reduce the potential destruction of intestinal epithelial cells by pathogenic bacteria [7,85]. With the increase in the distance between pathogenic bacteria and intestinal epithelial cells, the inflammatory response of animal intestines decreases [7,85]. On the other hand, intestinal stability is related to the health of animals. Especially for economic animals, the efficiency of nutrient absorption is affected by intestinal villi, which may be damaged by any environmental stress [88,89]. The addition of fermented fiber can be used to suppress the number of pathogenic bacteria or inflammation, thereby reduce the damage, and increase the length of intestinal villi [3,16].

In addition to the villus height, the motion of circular muscles in the intestine can drive the surface convection of the chyme. This movement is different from peristalsis and segmentation in mammals, a reflex type of motility [90]. The depth of the unstirred water layer is largely determined by the length of villi during the motion of the circular muscles which have various microbiota and can cause damage to the upper villus [89,90]. In some cases, such as in wheat or barley, which have a thicker unstirred water layer, there will be greater lumen viscosity and reduced oxygen transfer from mucosa to the intestinal cavity, thus increasing the activity of anaerobic bacteria [90,91]. However, the higher villus length can strengthen the agitation of chyme and alleviate the above situation.

### 7.2. Digestibility Adjustment in Animals

The quality of ingested feed is one of the most important environmental factors that affects animal production performance. Besides the energy density of feed and the composition of anti-nutritional factors, the digestibility of feed by animals is also an important indicator [82]. Higher digestibility can reduce the waste of nutrients in feed and reduce environmental pollution due to animal waste. Giving higher fiber can generally increase the digestive capacity of animals and can stabilize the composition of excreta more quickly [82]. However, excessive fiber addition (more than 40% peach palm (PP) meal replacement for maize in goats, the NDF corrected for ash and protein (NDFap) is 40.1% and acid detergent fiber (ADF) is 20.2% in 40% DM level of PP meal substitution group) may still cause a decrease in palatability, and the effect of digestibility is also related to the source of fiber [91]. Choi and Kim [82] and Navarro et al. [92] indicated out that although soluble fiber can be fermented in the intestine and produce short-chain fatty acids, insoluble fiber seems to increase the digestibility of animals.

Although intestinal health is closely related to the overall health of the animal, de Nanclares et al. [93] showed a positive correlation of the digestibility of pig with the enzyme activities but not with intestine morphology. Similarly, Liu et al. [94] pointed out that wheat bran has higher neutral detergent fiber (NDF), acid detergent fiber (ADF), and crude protein digestibility than soybean hull of animals, while exogenous addition of NSP-degrading enzyme could enhance nutrient utilization.

Chen et al. [58] also indicated that 1% addition of both soluble (inulin) and insoluble (lignocellulose) fiber could increase the digestibility of about 30% of dry matter, crude protein, and organic matter on weaning piglets. However, 5% cellulose, xylan, or β-glucan supplement was found to decrease the fecal digestibility of weaning pigs [65]. In growing pigs (21.3 ± 1.0 kg), 5% inulin supplement would decrease the digestibility of dry matter, NDF, and carbohydrates; however, the digestibility of ether extract increased. In contrast, although 5% carboxymethyl cellulose sodium supplement showed a similar result on the digestibility of ether extract compared to the inulin supplement group, the digestibility of other nutrients did not decrease, and even increased the NDF.

In the animal model of poultry, 2.5 or 5% supplement of oat hull, which is insoluble fiber, enhanced the gizzard weight and decreased the pH value in the intestine [83]. Fiber addition could further decrease the negative effects of pelleting and increase nutrient digestibility [83]. Likewise, Jiménez-Moreno et al. [95] also indicated that 2.5 or 5% supplementation with oat hull, rice hull, and sunflower hull did not decrease the growth performance in 21-day-old poultry. However, an increase in fiber supplements could improve water intake [95]. Overall, insoluble fiber can induce the digestibility of nutrition, thereby enhancing growth performance both in poultry and swine.

### 7.3. Fat Metabolism and Muscle Generation

In addition to promoting intestinal health, adding fiber can also reduce the amount or pattern of animal fat accumulation by promoting animal fat metabolism and reducing mRNA expression of animal fat synthesis genes [96]. Okrathok and Khempaka [62] showed that 1% cassava pulp could decrease the cholesterol content in the breast, thigh, liver, and serum in broilers. Similarly, in previous research reported by our team, we showed that addition of 0.5–2% mushroom waste compost could enhance adipolysis both in the liver and adipose tissues in broilers [10]. Moreover, the level of oxytocin, which promotes muscle formation, also increased in the 1–2% mushroom waste compost addition group compared to the control group by about 3.4 to 3.7-fold [10].

## 8. Fermented Fiber

### 8.1. Solid-State and Liquid Fermentations

During fermentation, microorganisms use specific substrates as energy sources to decompose, metabolize, or produce metabolites. Therefore, in general, the substrates fermented by microorganisms usually have bacterial proteins, secondary metabolites, small molecule peptides, and carbohydrates [97]. If the bacterial species used are GRAS (generally regarded as safe) probiotics and a nontoxic substrate, the fermentation products can be used as animal feed additives. According to different modes, fermentation can be either solid fermentation or liquid fermentation.

Solid-state fermentation uses a solid substrate, with adjusted moisture or pH, and probiotics are added to ferment at a specific temperature. The fermentation duration depends on different substrates or microorganisms [97]. Because of the low moisture content of solid-state fermentation, filamentous fungi or yeasts are generally considered suitable for growth [97]. In addition, microorganisms such as yeast or *Aspergillus oryzae*, with high activities of carbohydrate catalyzing enzymes are also used to ferment high-fiber substrates, such as wheat bran, straw, and lupin flour [17,25,98]. The high-protein soybean meal is suitable for fermentation using microorganisms with high protease activity such as *Lactobacillus* spp. and *Bacillus subtilis* [99]. After fermentation, cellulose and lignin content decrease, while hemicellulose and extractable functional metabolites increase slightly [3,53]. However, other crude ingredients, such as crude protein and minerals, are concentrated [53]. After probiotic fermentation, the anti-nutritional factors usually present in plants also decrease [97]. Fermented fiber also increases the number of probiotics in the intestinal tract and enhances the antioxidant capacity in animals. These characteristics increase the utility and benefits of fermented fibers [16,97].

In liquid fermentation, the water content in the fermentation substrate is much higher than the dry matter, or the solid substrate is directly immersed in the culture broth for fermentation [4]. The advantage of liquid fermentation is that the fermentation degree is relatively uniform and can easily adjust the nutrient composition and pH value in the culture broth while promoting the production of specific functional metabolites [4]. However, the functional components after fermentation are easily diluted by the culture solution and are not easily dried.

### 8.2. Stage Fermentation and Co-Fermentation

In recent years, the probiotic mode of fermentation has emerged. Because probiotics are not a panacea, a probiotic may only be good at secreting enzymes related to fiber or protein. Therefore, this problem can be solved through two-stage fermentation. Although there are only a few reports on two-stage fermentation for animal feed, this technology has been used in the brewing, biochemical, and decontamination industries for decades [100,101,102]. Two-stage fermentation can reduce the restrictions on the use of probiotics and also provide complementary effects. Scientists have also formulated fermented substrates with multiple probiotics at the same time; this could also enhance the quality of fermentation, although there may also be mutual inhibition effects.

Besides using probiotics for two-stage fermentation, enzymes and probiotics can also be used for co-fermentation [17]. Although probiotics can survive on many substrates, they may also encounter non-degradable anti-nutritional factors such as phytic acid, which is secreted in large amounts mainly by fungi or *E. coli* [50]. When probiotics ferment fibers, they release complex metabolites enveloped in fibers, including phytic acid and phenols [3,17]. When phytic acid sequesters minerals or amino acids, the growth rate of probiotics decreases [52]. However, when probiotics and phytase simultaneously ferment plant raw materials, probiotics can release phytic acid in the fiber, and phytase can further degrade phytic acid [17]. The released minerals or amino acids can further promote the growth of probiotics. The above cycle of co-fermentation of plant raw materials with probiotics and phytase will be more effective than using one of the substances alone [17].

Whether two-stage fermentation or co-fermentation can increase the availability of fiber needs to be assessed carefully. In addition to releasing the secondary metabolites of plants and reducing the content of anti-nutritional factors, the fermented fiber is also rich in probiotic metabolites, including bacterial proteins and short-chain fatty acids [31,103]. The bacterial protein can be used as one of the animal’s nutrient sources to promote animal growth. Short-chain fatty acids can lower the pH value in the intestine, maintain the health of intestinal epithelial cells, reduce inflammation, and provide animal energy [104]. Menconi et al. [105] also pointed out that the addition of organic acids can reduce the cost of producers and increase profits. A detailed discussion on the impact of short-chain fatty acids on animal health has been reported by Xu et al. [104].

## 9. Evaluation of the Use of Agricultural By-Products

Despite the mentioned benefits, the use of agricultural by-products, considering economic benefits or animal health, may still pose some risks and problems that must be addressed. Agricultural by-products are mostly rich in water. High-fiber and high-moisture substances are easily contaminated by mold and produce toxic mycotoxins [106], which decrease animal performance or even lead to death [107,108]. Therefore, the rapid drying of agricultural by-products is an important issue. However, because the agricultural by-products are not valued by producers, additional drying of agricultural by-products are not expected. To solve this problem, we put forward two suggestions. First, we encourage producers to discover and value the agricultural by-products and collect and dry them quickly after their production. We also suggest that agricultural by-products be collected quickly for adjusting water or pH, followed by fermentation with probiotics in a controlled environment to increase their utility value [4,16,17]. This approach will save the cost of drying the by-products once; however, it requires sufficient expertise to proceed.

On the other hand, because the agricultural by-product fiber structure is varied and complex, we still do not know the extent to which these composite fibers contribute to animal health. Considering that not every fiber may be suitable for improving animal health, the composition, and content of fiber affect the composition of intestinal microbiota, and intestinal microbes are closely related to virus composition in the animal gut [21,109]. Therefore, we suggest an integrated assessment of the effects of different agricultural by-products on animal health and their correlations through emerging metabolomics, microbiome, and virome analysis [7,21,109,110].

## 10. Conclusions

Although the world’s food demand is rising, people are pursuing a better quality of life while also noticing that the earth’s environment needs to be protected. Food is a rigid requirement of life. Since the production of agricultural by-products is inevitable, we expect their use as animal feed or feed additives after appropriate treatments. Studies in recent years have repeatedly pointed out that fiber as an animal feed or feed additive does not seem to be so unfeasible as previously thought. Instead, dietary fiber and other functional compounds, such as polyphenol and flavonoids, could enhance health, antioxidant capacities, and stabilize the microbiota in animals. In addition, agricultural by-products are a suitable and inexpensive source of fiber; they are not only inexpensive, but their proper use can also reduce costs of waste disposal and animal feeding. Scientists must integrate metabolomics, microbiome, and virome analysis to jointly evaluate the effects of different fiber compositions on animals. Overall, we recommend that animal producers be encouraged to use high-fiber agricultural by-products as animal feed and feed additives.

## Figures and Tables

**Figure 1 animals-11-02098-f001:**
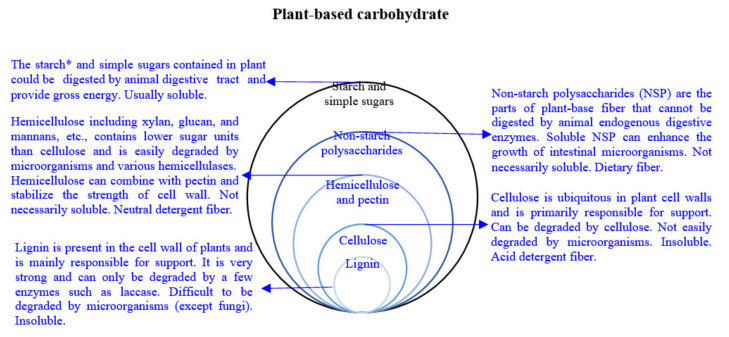
The classification and characteristics of carbohydrates in plants. * Resistant starch is beyond the scope of this paper.

**Table 1 animals-11-02098-t001:** Effect of different fiber replacement or addition on animal growth performance and the microbes in the ileum.

Animal Type	Ingredient	Age	Effect ^1^			References
			ADG	FCR	Ileum Microbes (log CFU)	
Ross 308 broilers	10% wheat bran replace	1–35 days	−4%	+1%	Coliform: −1.53 *Clostridium perfringens*: +0.08	[61]
	10% Trichoderma fermented wheat bran replace		−1%	−6%	Coliform: −0.6 *C. perfringens*: −0.26	
Ross 308 broilers	3% pectin	1–14 days	−14%	+21%	*E. coli*: +0.97 * *Lactobacillus* spp.: −1.57 *	[14]
	3% carboxymethyl cellulose		−14%	+21%	*E. coli*: +1.54 * *Lactobacillus* spp.: −1.72 *	
	3% cellulose		−3%	+5%	*E. coli*: +0.15 *Lactobacillus* spp.: +0.1	
Ross 308 broilers	0.5% cassava pulp addition	1–42 days	+0%	−1%	-	[62]
	1% cassava pulp addition		+0%	−2%	-	
	1.5% cassava pulp addition		+1%	−1%	-	
Ross 308 broilers	5% wheat bran replace	1–35 days	+2%	−1%	Coliform: −0.12 Lactic acid bacteria: +0.16	[63]
	10% wheat bran replace		−3%	+1%	Coliform: +0.16 Lactic acid bacteria: +0.2	
	5% *Laetiporus sulphureus* fermented wheat bran replace		+2%	−3% *	Coliform: −0.68 Lactic acid bacteria: +0.37	
	10% *Laetiporus sulphureus* fermented wheat bran replace		+1%	−3% *	Coliform: −0.29 Lactic acid bacteria: +0.47	
Weaning piglets	1% insoluble fiber	24–52 days	+3%	−3%	*E. coli*: +0.38 *Lactobacillus* spp.: +0.9 *	[64]
	1% soluble fiber		−3%	−1%	*E. coli*: +0.22 *Lactobacillus* spp.: +0.57	
	CRMDF ^2^		+11%	−8% *	*E. coli*: −0.36 *Lactobacillus* spp.: +0.53	
	0.5% insoluble fiber and 0.5% soluble fiber		+6%	−7% *	*E. coli*: +0.61 *Lactobacillus* spp.: +0.95	
Weaning piglets	5% cellulose	21–46 days	−19%	+9%	-	[65]
	5% xylan		−15%	+0%	-	
	5% glucan		−22% *	+3%	-	

^1^ Increase in average daily weight gain, feed conversion rate, and microbe numbers (log CFU) were compared to those of the control group; ADG: average daily weight gain; FCR: feed conversion rate. ^2^ CRMDF: 0.75% insoluble fiber (lignocellulose) +0.25% soluble fiber (insulin) diet during the first two weeks and 0.25% insoluble fiber +0.75% soluble fiber diet during the last two weeks (totally 4 weeks after weaned). * Indicates a significant difference between treatment and control groups.

**Table 2 animals-11-02098-t002:** Effect of fiber supplements on digestibility, health, and animal production performance.

Animal Type	Ingredient	Periods	Effects	References
Goat	10, 40, 60, and 85% palm meal replace corn	90–188 day	Reduce the feed intake, apparent digestibility, and palatability on higher replacement	[84]
Weaning pig	1% inulin or lignocellulose addition	24–52 day	Increase apparent digestibility of ileum and tight junction expression	[64]
Growing pig	5% inulin	21.3 ± 1.0 kg	Decrease the dry matter digestibility of ileum, neutral and carbohydrates; increase the ether extract digestibility and total short-chain fatty acids in feces	[81]
	5% carboxymethyl cellulose sodium		Increase the digestibility of ileum, detergent fiber, and ether extract	
Sow	40 ppm chitooligosaccharide	Production and lactation	Increase litter number, litter weight, and survival rate	[67]
Barrows	High-fiber treatment ^1^	81.5 kg	Increase total tract digestibility of gross energy, dry matter, organic matter, and crude protein	[82]
Weaning piglets	5% cellulose	21–46 day	Decrease fecal digestibility of dry matter, calcium, phosphorus, energy, and crude fiber	[65]
	5% xylan		Decrease fecal digestibility of calcium	
	5% glucan		Decrease fecal digestibility of calcium, phosphorus, and crude fiber; enhance the gut barrier function	
Cobb-500 broilers	2.5 or 5% oat hulls supplement	1–21 day	Enhance the gizzard weight and decrease the pH value in the intestine, increase nutrient digestibility	[83]
Ross 308 broilers	0.5, 1, and 1.5% cassava pulp addition	1–42 day	Decrease the cholestenone concentration in liver, serum, and muscle, increase nutrition digestibility and gizzard weight	[62]
Ross 308 broilers	0.5, 1, and 2% PWMC ^2^	1–35 day	Increase antioxidant capacities, tight junction expression, and enhance fat metabolism	[3]
Ross 308 broilers	5% wheat bran supplement	1–35 day	Increase IL-6 and IL-1β mRNA expression	[63]
	5% Laetiporus sulphureus fermented wheat bran supplement		Increase IgA secretion in serum and ileum; decrease IL-1β and TNF-α concentration in serum	

^1^ High-fiber treatment: 296 g/kg amylase-treated neutral detergent fiber (aNDF) and 113 g/kg acid detergent fiber (ADF) compared to 50 g/kg aNDF and 16 g/kg ADF. ^2^ PWMC: *Pennisetum purpureum* schum No. 2 waste mushroom compost.

## Data Availability

The dataset supporting this study is present within the article.

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
