# Peer review of "The Potential Utilization of High-Fiber Agricultural By-Products as Monogastric Animal Feed and Feed Additives: A Review"

_animals, 2021, doi:10.3390/ani11072098_

Round 1

Reviewer 1 Report

The authors introduced the composition, function and effect of fiber in agricultural by-product. The manuscript was well organized and written. Some grammar mistakes as follows should be revised.

Line 85-86:Corn grains only account for about 20% and soybeans account for about 30% of the dry whole corn plant. Why does the corn plant have soybean?

Line 86-87: Accordingly, the residues are burn out or as a material for compost not only exacerbating air pollution but also causing waste of resources

In Figure 1: The starch and simple sugars contained in plant could be digested by animal digestive tract and provide gross energy.

Line 188-189: Cumulatively, the composition and amount of tannic acid to assess its effects on animal health. What is the meaning of whole sentence?

Delete “The” in front of effect in title of Table 1 and 2. All items should be single. “Affect” in Table 1 should be “effect”.

Delete “The” in the beginning of all sub-titles and titles of table or figure.

Line 377-379” Although intestinal health is closely related to the overall health of the animal, de? Nanclares et al. [93] showed a positive correlation of the digestibility of pig with the enzyme activities but not with intestine morphology.

Author Response

The authors appreciate the comments from the reviewers. The manuscript has been revised in accordance with their requests. We have tried our best to take all comments into account, incorporating them into the revised manuscript as indicated in our responses to the reviewer.

(Revisions related to reviewer’s comments are shown in blue in the revised manuscript)

Responses to Reviewer I 's comments:

The authors introduced the composition, function and effect of fiber in agricultural by-product. The manuscript was well organized and written. Some grammar mistakes as follows should be revised.

Response: We really appreciate the comments, and thank you for providing the useful suggestion.

  1. Line 85-86:Corn grains only account for about 20% and soybeans account for about 30% of the dry whole corn plant. Why does the corn plant have soybean?
    Response: Thank you for the suggestion, we have corrected the sentence as “Corn grains only account for about 20% [18] and soybeans account for about 30% of the dry whole plants.” in the Line 85-86.

  2. Line 86-87: Accordingly, the residues are burn out or as a material for compost not only exacerbatingair pollution but also causing waste of resources
    Response: Thanks for the suggestion, we have corrected the sentence as “Accordingly, the residues are burn out or as a material for compost not only exacerbating air pollution but also causing waste of resources.” in the Line 86-87.

  3. In Figure 1: The starch and simple sugars containedin plant could be digested by animal digestive tract and provide gross energy.
    Response: Thanks for the suggestion, we have corrected the sentence as “The starch and simple sugars contained in plant could be digested by animal digestive tract and provide gross energy.” in the Figure 1.

  4. Line 188-189: Cumulatively, the composition and amount of tannic acid to assess its effects on animal health. What is the meaning of whole sentence?
    Response: Thanks for the suggestion, we have corrected the sentence as “Therefore, the appropriate amount of tannic acid and animal trial for evaluation, it will has a positive effect on animal health” in the Line 188-189.

  5. Delete “The” in front of effect in title of Table 1 and 2. All items should be single. “Affect” in Table 1 should be “effect”.
    Response: Thanks for the suggestion, we have corrected the sentences as reviewer suggestion.

  6. Delete “The” in the beginning of all sub-titles and titles of table or figure.
    Response: Thanks for the suggestion, we have corrected the sentences as reviewer suggestion.

  7. Line 377-379” Although intestinal health is closely related to the overall health of the animal, de?Nanclares et al. [93] showed a positive correlation of the digestibility of pig with the enzyme activities but not with intestine morphology.
    Response: Thanks for the suggestion, we have deleted the “de” in the Line 377-379.

Reviewer 2 Report

The subject of the paper is very interesting and  currently in the center of interest of many researchers as well in accordance with commonly implanting “green policy”. However, in my opinion the paper requires major changes and improvements especially in paper structure and comprehension.  Moreover authors many times use active forms of the sentences (Line 232, 233, 319, 323 etc.) while recommended for scientific papers is passive form.

The title of the paper The potential utilization of high-fiber agricultural by-products 2 as monogastric animal feed additives: A review indicates that the subject of the article are high-fiber feed additives. At the same time in the paper authors discussed various feed materials such as brans, meals and other by-products that are not the feed additives but feed materials or for instance inulin that obviously is the well-common feed additive but rather not by-product…

The simple summary in my opinion is too short and do not inform about the paper content.

Abstract

Line 19: “…polyphenols and flavonoids…” Flavonoids are polyphenols, aren’t they?

Line 27:  “.. value of agricultural by-products….” What value?

  1. Introduction

Line 49-51: I don’t understand the sentences  “the second category …..” It should be clarified.

Line 51: Economic animal breeders? Who are they? Actually all animal producers want to produce at high level and high quality products with as much as it possible low production costs. Also I do not think that fiber can be treated as costs reduce factor?

  1. The potential utilization of feed crop by-products

Line 85-86: “Corn grains only account for about 20% [18] and soybeans account for about 30% of the dry whole corn plant.” Soybean in whole corn plant?

Line 87: “are burn out”. Should be are burnt out.

  1. Composition of fiber

Line 123: “affects” – should be effects

  1. 4. Phytochemicals and anti-nutrition factors in plants

Generally, the description of individual biologically active compounds in the dietary fiber is to shallow. There is a lack of schemes/mechanism of actions

Line 134: “…., and lectins are also anti-nutritional…” should be “…., and lectins, there are also anti-nutritional…”

Line 181-182: “Not only in poultry, but tannic acid also reduces plasma iron levels and the performance of weaned piglets [45].” I suggest Tannic acid also reduces plasma iron levels and the performance of weaned piglets [45].

Line 188-189: “Cumulatively, the composition and amount of 188 tannic acid to assess its effects on animal health.” This sentence doesn’t make sense.

  1. The effect of fiber addition on animal production performance and microbiota

Table 1. What is it CRMDF? Please explain the abbreviation.

The description of the sub-chapters are too short and not enough detailed. The first part of 5.2. “Pigs have different nutritional requirements in different stages of their life cycle. 253 For piglets during lactation, sow milk is the most important source of energy, therefore, 254 it is important to maintain stable production of sow milk and ensuring that every piglet 255 consumes milk [67]. Piglets face huge environmental changes and stress when weaning, 256 while for the piglets that have just been weaned, there are significant changes in the type 257 of food ingested, so diarrhea is prone to occur. Hence, it is necessary to maintain stable 258 feeding and improve the physiological health of pigs during their late growth period or 259 during sow pregnancy.” do not provides any essential knowledge to the subject of sub-chapter.

  1. Effect of functional components of fibers on animal physiology

Line 277-279: Wrong used references. “Lee et al have discussed in detail the inflammation, antioxidant mechanism and 277 phytogenics, and fungus type probiotics on animal immune regulation and antioxidant 278 capacity [69,70,10].”

6.1. Anti-oxidant and anti-inflammatory responses

Actually only the  last two sentence touch the core of the sub-chapter. The description is to shallow and does not introduce any essential knowledge on influence of fiber on Anti-oxidant and anti-inflammatory responses.

Line 312-316: For sure it is not related to Effect of functional components of fibers on animal physiology.

  1. Evaluation of the use of agriculture by-products

Line 479:  “´- should be or even death or even lead to death

The paper raises important issues regarding possibility of inclusion of agro-food by-products in monogastric animals. It is a valuable bridgehead to knowledge in this area. However, I suggest rethinking the work layout and editing the work. In my opinion many of description included in the paper are too general and do not explain physiological mechanisms of compounds included in fibers on animal production and health.

Author Response

The authors appreciate the comments from the reviewers. The manuscript has been revised in accordance with their requests. We have tried our best to take all comments into account, incorporating them into the revised manuscript as indicated in our responses to the reviewer.

(Revisions related to reviewer’s comments are shown in blue in the revised manuscript)

Responses to Reviewer II 's comments:

The subject of the paper is very interesting and currently in the center of interest of many researchers as well in accordance with commonly implanting “green policy”. However, in my opinion the paper requires major changes and improvements especially in paper structure and comprehension.  Moreover authors many times use active forms of the sentences (Line 232, 233, 319, 323 etc.) while recommended for scientific papers is passive form.

Response: We really appreciate the comments, and thank you for providing the useful suggestion. We have corrected the sentences (Line 232, 233, 319, 323 etc.) as reviewer suggestion.

  1. The title of the paper The potential utilization of high-fiber agricultural by-products as monogastric animal feed additives: A review indicates that the subject of the article are high-fiber feed additives. At the same time in the paper authors discussed various feed materials such as brans, meals and other by-products that are not the feed additives but feed materials or for instance inulin that obviously is the well-common feed additive but rather not by-product…
    Response: Thanks for the suggestion, we have corrected the title of the paper “The potential utilization of high-fiber agricultural by-products as monogastric animal feed and feed additives: A review”. Otherwise, mention insulin mainly based on high-fiber agricultural by-products include the fermentability of fiber is related to the microbial composition of the intestine and short-chain fatty acids, while the solubility is related to the satiety and fecal configuration of animals [11,12]. Insoluble fibers generally comprise cellulose and lignin, while soluble fibers include inulin and pectin [12]. Interestingly, most soluble fibers are more fermentable than insoluble fibers. Therefore, soluble fiber may also promote excessive growth of harmful microorganisms and increase chyme viscosity [14]. In this review, it is mentioned that inulin is included in the application research of dietary fiber in animals.
    In total, simultaneous addition of 1% soluble (inulin) and insoluble (lignocellulose) fiber could significantly increase the feed conversion rate of weaning pigs (24 to 52-day-old); however, adding soluble fibers or insoluble fibers singly is less effective [64].
    2. Chen et al. [58] also indicated that both soluble (inulin) or insoluble (lignocellulose) fiber 1% addition could increase the digestibility of about 30% of dry matter, crude protein, and organic matter on weaning piglets.

  2. The simple summary in my opinion is too short and do not inform about the paper content.
    Response: Thanks for the suggestion. We newly added the description as “High fiber agriculture by-products, which can enhance animal performance and health, have the potential to be used as feed additives. Before using high fiber agriculture by-products, it is necessary to pay attention to the problem of anti-nutritional factors and contamination due to mycotoxins. Solubility and fermentability are the keys that mainly affect fiber availability. Recent years have pointed out that fiber as an animal feed or feed additive does not seem to be so unfeasible as previously thought. Instead, dietary fiber and other functional compounds, such as polyphenol and flavonoids, could enhance health, anti-oxidant capacities, and stabilize the microbiota in animals. In addition, high fiber agriculture by-products are a suitable and inexpensive source of fiber, they proper use may reduce costs of animal feeding. Scientists must integrate characteristics and appropriate usage analysis to jointly evaluate the effects of different fiber compositions on those animals. Based on this foundation, animal products be encouraged to use high-fiber agricultural by-products as animal feed and feed additives.” In the simple summary

Abstract

  1. Line 19: “…polyphenols and flavonoids…” Flavonoids are polyphenols, aren’t they?
    Response: Thanks for the valuable suggestion. We have corrected as “functional metabolites such as polyphenol (included flavonoids), that can promote animal health.”

  2. Line 27:  “.. value of agricultural by-products….” What value?
    Response: Thanks for the suggestion. We have corrected as “utilization value” in Abstract.

Introduction

  1. Line 49-51: I don’t understand the sentences  “the second category …..” It should be clarified.
    Response: Thanks for the suggestion. We have corrected the sentence as “The second category is a dietary fiber cellulose, chitin and other fibers which may ferment by the microorganism.”
  2. Line 51: Economic animal breeders? Who are they? Actually all animal producers want to produce at high level and high quality products with as much as it possible low production costs. Also I do not think that fiber can be treated as costs reduce factor?
    Response: Thanks for the valuable suggestion. We newly corrected the sentence as “The second category is a dietary fiber cellulose, chitin and other fibers which may ferment by the microorganism. In the past, scientists and animal producers considered that the role of dietary fiber in feed is mainly to dilute protein or energy [5]”.

The potential utilization of feed crop by-products

  1. Line 85-86: “Corn grains only account for about 20% [18] and soybeans account for about 30%of the dry whole corn plant.” Soybean in whole corn plant?
    Response: Thanks for the suggestion. We have corrected the sentence as “Corn grains only account for about 20% [18] and soybeans account for about 30% of the dry whole plants.”
  2. Line 87: “are burn out”. Should be are burnt out.
    Response: Thanks for the suggestion. We have corrected as “burnt out”.

Composition of fiber

  1. Line 123: “affects” – should be effects
    Response: Thanks for the suggestion. We have corrected as “effects”.

Phytochemicals and anti-nutrition factors in plants

  1. Generally, the description of individual biologically active compounds in the dietary fiber is to shallow. There is a lack of schemes/mechanism of actions
    Response: Thanks for the suggestion. We newly added the description as “In addition to fiber, plants also contain a variety of secondary metabolites, including antioxidant peptides, phenols, flavonoids, phytic acid, trypsin inhibitors, and lectins [22,23]. These plant-derived secondary metabolites are rich in antioxidant, anti-inflammatory, and antibacterial activity, but may also reduce the animal’s absorption efficiency to nutrients such as minerals or amino acids [22]. Phytochemicals have been shown to positive antioxidant benefits toward animals in terms of favored growth performance, production quality, and enhanced endogenous antioxidant system, possibly by directly affecting specific molecular targets or/and indirectly as stabilized conjugates affecting the metabolic pathways [11,13,23]. Accordingly, dissecting the antioxidant effects and the underlying mechanism of dietary phytochemicals is an important area. Therefore much of attention is being focused on a new wave of nutrigenomics [6,10,23]. Several studies dedicated to understand and formulate mechanistic pathways by which these naturally-derived substances could alter the fate of cells, particularly the antioxidant properties of phytochemicals have been implicated as stress-alleviation agents [11,23,24]. Besides, phytic acid, trypsin inhibitors, and lectins, there are also anti-nutritional factors commonly found in plants [24,25] and cannot be ignored while using high-fiber agricultural by-products as animal feed or feed additive.” in the section.

  2. Line 134: “…., and lectins are also anti-nutritional…” should be “…., and lectins, there are also anti-nutritional…”
    Response: Thanks for the suggestion. We have corrected the sentence as “Besides, phytic acid, trypsin inhibitors, and lectins, there are also anti-nutritional factors commonly found in plants [24,25] and cannot be ignored while using high-fiber agricultural by-products as animal feed.”

  3. Line 181-182: “Not only in poultry, but tannic acid also reduces plasma iron levels and the performance of weaned piglets [45].” I suggest Tannic acid also reduces plasma iron levels and the performance of weaned piglets [45].
    Response: Thanks for the suggestion. We have corrected the sentence as “Tannic acid also reduces plasma iron levels and the performance of weaned piglets [45].”

  4. Line 188-189: “Cumulatively, the composition and amount of tannic acid to assess its effects on animal health.” This sentence doesn’t make sense.
    Response: Thanks for the suggestion. We have corrected the sentence as “Therefore, the appropriate amount of tannic acid and animal trial for evaluation, it will has a positive effect on animal health.”

The effect of fiber addition on animal production performance and microbiota

  1. Table 1. What is it CRMDF? Please explain the abbreviation.
    Response: Thanks for the suggestion. CRMDF is one group of treatments for diet fiber supplementation strategy for the weaned piglets totally 4 weeks trial. The CRMDF group use the strategy as 0.75% insoluble fiber (lignocellulose) + 0.25% soluble fiber (insulin) diet during the first two weeks and 0.25% insoluble fiber + 0.75% soluble fiber diet during the last two weeks.

(64) Chen, T.; Chen, D.; Tian, G.; Zheng, P.; Mao, X.; Yu, J.; He, J.; Huang, Z.; Luo, Y.; Luo, J.; Yu, B. Effects of soluble and insoluble dietary fiber supplementation on growth performance, nutrient digestibility, intestinal microbe and barrier function in weaning piglet. Anim. Feed Sci. Tech. 2020, 260, 114335.

  1. The description of the sub-chapters are too short and not enough detailed. The first part of 5.2. “Pigs have different nutritional requirements in different stages of their life cycle. For piglets during lactation, sow milk is the most important source of energy, therefore, it is important to maintain stable production of sow milk and ensuring that every piglet consumes milk [67]. Piglets face huge environmental changes and stress when weaning, while for the piglets that have just been weaned, there are significant changes in the type of food ingested, so diarrhea is prone to occur. Hence, it is necessary to maintain stable feeding and improve the physiological health of pigs during their late growth period or during sow pregnancy.” do not provides any essential knowledge to the subject of sub-chapter.
    Response: Thanks for the suggestion. We have added and corrected the sentence as “Pigs have different nutritional requirements in different stages of their life cycle. For piglets during lactation, sow milk is the most important source of energy, therefore, it is important to maintain stable production of sow milk and ensuring that every piglet consumes milk [65,67]. Piglets face huge environmental changes and stress when weaning, while for the piglets that have just been weaned, there are significant changes in the type of food ingested, so diarrhea is prone to occur [45,64]. Feeding piglets after weaning provides an appropriate amount of insoluble fiber (as inert fiber) to avoid diarrhea caused by accumulation of undigested nutrients and help piglets to restore intestinal function [64,65]. Moreover, it is necessary to maintain stable feeding and improve the physiological health of pigs during their late growth period or during sow pregnancy. Therefore, adding dietary fiber to the sow diet could increase feed intake during lactation, thereby increasing the number of weaned piglets, weaned piglet weight and average daily gain. Feeding high-fiber diets in late pregnancy may also reduce the stillbirth rate by reducing delivery time. In addition, it can alleviate constipation in pregnant and lactating sows [67,68].

Effect of functional components of fibers on animal physiology

  1. Line 277-279: Wrong used references. “Lee et al have discussed in detail the inflammation, antioxidant mechanism and phytogenics, and fungus type probiotics on animal immune regulation and antioxidant capacity [69,70,10].”
    Response: Thanks for the suggestion. We have corrected the sentence as “Previous studies”.

6.1. Anti-oxidant and anti-inflammatory responses

  1. Actually only the last two sentence touch the core of the sub-chapter. The description is to shallow and does not introduce any essential knowledge on influence of fiber on Anti-oxidant and anti-inflammatory responses.
    Response: Thanks for the suggestion.

Previous studies have discussed in detail the inflammation and antioxidant mechanism of phytochemicals on animal immune regulation and antioxidant capacity [69,70,10]. Briefly, when animals are subjected to environmental stress, such as heat stress or pathogens infection, stimulation causes an inflammatory response and increases oxidative stress in animals [71]. Stimulated by pathogenic bacteria, the animal initiates an immune response leading to a cytokine storm [71]. However, excessive inflammation can reduce animal performance and even lead to death [10]. On the other hand, when the oxidative pressure is too high, animals are not able to eliminate the damage caused by free radicals to cells or organs [69]. Among them, the animal's antioxidant system is mainly regulated by the liver, so the antioxidant capacity is also related to liver performance [3]. Applying phytochemicals or botanical compounds to the feed could promote intestinal health, reduce the inflammatory response and enhance the antioxidant capacity of the animal [10]. The mechanisms are majorly the nuclear factor (erythroid-derived 2)-like 2 (Nrf2) and nuclear factor kappa B (NF-κB) for they are respectively the key transcription factors involved in oxidative stress and inflammation for elucidating the underlying signal transduction pathways. Therefore, phytochemicals can regulate these transcription factors leading to the improvement of oxidative status, heme oxygenase-1 (HO-1) gene is found crucial for Nrf2-mediated NF-κB inhibition. Hence, proper fibers as phytochemicals (likely 0.5-1% mulberry leaves addition in laying hens) with such modulatory effects worth be used to explore the possible crosstalk in oxidative stress and immunomodulation in animals [69,70].

  1. Line 312-316: For sure it is not related to Effect of functional components of fibers on animal physiology.
    Response: Thanks for the suggestion. We have deleted the not related sentences as reviewer reminding.

Evaluation of the use of agriculture by-products

  1. Line 479:  “´- should be or even death or even leadto death
    Response: Thanks for the suggestion. We have corrected the sentence as “even lead to death”.

The paper raises important issues regarding possibility of inclusion of agro-food by-products in monogastric animals. It is a valuable bridgehead to knowledge in this area. However, I suggest rethinking the work layout and editing the work. In my opinion many of description included in the paper are too general and do not explain physiological mechanisms of compounds included in fibers on animal production and health.
Response: We really appreciate this comment and the suggestion. We have newly supplemented and corrected the sections as above response and revised manuscript to explain physiological mechanisms of compounds included in fibers on animal production and health. Moreover, many thanks for your suggestion to help the reader getting my point. Thanks a lot again.

Reviewer 3 Report

Dear Authors,

The manuscript under review concerns the potential use of high-fibre agricultural by-products as feed additives for monogastric animals. The paper is interesting, although in my opinion, those passages that directly provide an answer to the subject of the paper need to be strengthened. Here are some remarks that came to mind while checking the manuscript:

Line 43: please move a part of a sentence from line 44 to line 43, an error occurred
Line 80: What is the novelty of the manuscript submitted for appraisal in relation to other works on similar topics? It is worth highlighting this for the purpose of the paper
Line 106: What is the source of figure 1? Is it an own study based on a specific source? This should be indicated. 
Line 236: in Table 1 change hyphen to minus sign. In addition, it is necessary to 
edit Table 1 - the literature entries are not all centered
Line 44: change the author font from the literature item numbered 14.
Line 277: A full stop is missing when citing Lee et al., please correct.
Line 291: please consider another title for this subchapter
Line 317: A question mark is missing from the title of the subsection, please complete.
Line 327: Please edit the table correctly, no centering of text in each cell the same.
Line 527: Delete the letter a from the 2020a notation - there is no need when quoting different authors of publications from a given year.

Author Response

The authors appreciate the comments from the reviewers. The manuscript has been revised in accordance with their requests. We have tried our best to take all comments into account, incorporating them into the revised manuscript as indicated in our responses to the reviewer.

(Revisions related to reviewer’s comments are shown in blue in the revised manuscript)

Responses to Reviewer III 's comments:

The manuscript under review concerns the potential use of high-fibre agricultural by-products as feed additives for monogastric animals. The paper is interesting, although in my opinion, those passages that directly provide an answer to the subject of the paper need to be strengthened. Here are some remarks that came to mind while checking the manuscript:

Response: We really appreciate the comments, and thank you for providing the useful suggestion.

  1. Line 43: please move a part of a sentence from line 44 to line 43, an error occurred
    Response: Thanks for the suggestion. We newly added the sentence as “Agricultural by-products from different plant sources may also contain dietary fiber (such as cellulose and hemicellulose), starch, crude protein, oligosaccharides or vitamins, and other nutrients [1,3]. Phenols and flavonoids, the most important phytochemicals, are often included in the fiber [3,4].” In the Introduction

  2. Line 80: What is the novelty of the manuscript submitted for appraisal in relation to other works on similar topics? It is worth highlighting this for the purpose of the paper
    Response: Thanks for the suggestion. Base on reviewer concerning, we more emphasize the use of by-product fiber (including functional metabolites) and its impact on animal health and discuss its regulation mechanism on animal physiology. We newly described the section of Simple Summary, Phytochemicals and anti-nutrition factors in plants and Effects of fiber on swine sections as:
  • High fiber agriculture by-products, which can enhance animal performance and health, have the potential to be used as feed additives. Before using high fiber agriculture by-products, it is necessary to pay attention to the problem of anti-nutritional factors and contamination due to mycotoxins. Solubility and fermentability are the keys that mainly affect fiber availability. Recent years have pointed out that fiber as an animal feed or feed additive does not seem to be so unfeasible as previously thought. Instead, dietary fiber and other functional compounds, such as polyphenol and flavonoids, could enhance health, anti-oxidant capacities, and stabilize the microbiota in animals. In addition, high fiber agriculture by-products are a suitable and inexpensive source of fiber, they proper use may reduce costs of animal feeding. Scientists must integrate characteristics and appropriate usage analysis to jointly evaluate the effects of different fiber compositions on those animals. Based on this foundation, animal products be encouraged to use high-fiber agricultural by-products as animal feed and feed additives.
  • In addition to fiber, plants also contain a variety of secondary metabolites, including antioxidant peptides, phenols, flavonoids, phytic acid, trypsin inhibitors, and lectins [22,23]. These plant-derived secondary metabolites are rich in antioxidant, anti-inflammatory, and antibacterial activity, but may also reduce the animal’s absorption efficiency to nutrients such as minerals or amino acids [22]. Phytochemicals have been shown to positive antioxidant benefits toward animals in terms of favored growth performance, production quality, and enhanced endogenous antioxidant system, possibly by directly affecting specific molecular targets or/and indirectly as stabilized conjugates affecting the metabolic pathways [11,13,23]. Accordingly, dissecting the antioxidant effects and the underlying mechanism of dietary phytochemicals is an important area. Therefore much of attention is being focused on a new wave of nutrigenomics [6,10,23]. Several studies dedicated to understand and formulate mechanistic pathways by which these naturally-derived substances could alter the fate of cells, particularly the antioxidant properties of phytochemicals have been implicated as stress-alleviation agents [11,23,24]. Besides, phytic acid, trypsin inhibitors, and lectins, there are also anti-nutritional factors commonly found in plants [24,25] and cannot be ignored while using high-fiber agricultural by-products as animal feed or feed additive.
  • Pigs have different nutritional requirements in different stages of their life cycle. For piglets during lactation, sow milk is the most important source of energy, therefore, it is important to maintain stable production of sow milk and ensuring that every piglet consumes milk [65,67]. Piglets face huge environmental changes and stress when weaning, while for the piglets that have just been weaned, there are significant changes in the type of food ingested, so diarrhea is prone to occur [45,64]. Feeding piglets after weaning provides an appropriate amount of insoluble fiber (as inert fiber) to avoid diarrhea caused by accumulation                                                                                                                                                                                                                                                                                                                                                                                                                                                                                                                                                                                                                                                                                                                                                                                                                                                                                                                                                                                                                                                                                                                                                                                                                                                                                                                                                                                                                         of undigested nutrients and help piglets to restore intestinal function [64,65]. Moreover, it is necessary to maintain stable feeding and improve the physiological health of pigs during their late growth period or during sow pregnancy. Therefore, adding dietary fiber to the sow diet could increase feed intake during lactation, thereby increasing the number of weaned piglets, weaned piglet weight and average daily gain. Feeding high-fiber diets in late pregnancy may also reduce the stillbirth rate by reducing delivery time. In addition, it can alleviate constipation in pregnant and lactating sows [67,68].

  1. Line 106: What is the source of figure 1? Is it an own study based on a specific source? This should be indicated.
    Response: Thanks for the suggestion. We have corrected and added the description as “The figure information adapted from Prasad and Bondt [12] and International Scientific Association for Probiotics and Prebiotics (ISAPP) organization [19].” in the Figure 1.

  2. Line 236: in Table 1 change hyphen to minus sign. In addition, it is necessary to
    Response: Thanks for the suggestion. We have changed hyphen to minus sign.

  3. edit Table 1 - the literature entries are not all centered
    Response: Thanks for the suggestion. We have corrected them.

  4. Line 44: change the author font from the literature item numbered 14.
    Response: Thanks for the suggestion. We have corrected it.

  5. Line 277: A full stop is missing when citing Lee et al., please correct.
    Response: Thanks for the suggestion. We have corrected them as “Previous studies have discussed in detail the inflammation and antioxidant mechanism of phytochemicals on animal immune regulation and antioxidant capacity [69,70,10].”.

  6. Line 291: please consider another title for this subchapter
    Response: Thanks for the suggestion. We have corrected them as Fibers satiety in animals” for this subchapter.

  7. Line 317: A question mark is missing from the title of the subsection, please complete.
    Response: We really appreciate this comment and the suggestion. Considered to fit the content description, the title of the subsection modified to “Fibers can increase animal performances”.

  8. Line 327: Please edit the table correctly, no centering of text in each cell the same.
    Response: Thanks for the suggestion. We have edit the table correctly.

  9. Line 527: Delete the letter a from the 2020a notation - there is no need when quoting different authors of publications from a given year.
    Response: Thanks for the suggestion. We have deleted it.

Reviewer 4 Report

Dear authors,

I have comments and corrections marked in the text for you. Please make them.

Author Response

The authors appreciate the comments from the reviewers. The manuscript has been revised in accordance with their requests. We have tried our best to take all comments into account, incorporating them into the revised manuscript as indicated in our responses to the reviewer.

(Revisions related to reviewer’s comments are shown in blue in the revised manuscript)

Responses to Reviewer IV 's comments:

Dear authors,

I have comments and corrections marked in the text for you. Please make them.
Response: We really appreciate the comments and the suggestion.

  1. Expand SCFA
    Response: We really appreciate the suggestion. We have corrected the description as “short-chain fatty acids (SCFA).” in the Introduction.

  2. microbiota
    Response: Thanks for the suggestion. We have corrected the description as “microbiota”.

  3. Not all fibers do this. It is necessary to point out

Response: We really appreciate the suggestion. We have corrected the description as “Fiber, like Pennisetum grass, a kind of common forage species, can also be used as a medium for mushroom cultivation [3,10]. Previous study showed those mushroom wastes compost (high-fiber agricultural by-products) can reduce the inflammatory response and increase the antioxidant capacity of serum and liver of animals, and enhance the growth of intestinal villi when adjusted added in animal’s diet [3,6,10]. Moreover, adding dietary fiber (as potato pulp, sugar beet pulp and pectin residue) can also promote the gastric emptying rate and alter the satiety of animals, the authors results can also help sows maintain the shape of feces and reduce the incidence of diarrhea by water retaining contents [11].

  1. Give some examples
    Response: Thanks for the suggestion. We have added the description as “In vitro, fermentation of fibers with probiotics (Saccharomyces cerevisiae and Aspergillus oryzae) and/or enzymes (phytase) can form postbiotics with fermentation products”.

  1. it is necessary to support this paragraph with references
    Response: Thanks for the suggestion. We have corrected the description and reference as “The International Scientific Association for Probiotics and Prebiotics (ISAPP) defines prebiotic as “a substrate that is selectively utilized by host microorganisms conferring a health benefit [19].”
    [19] Hill, C.; Guarner, F.; Reid, G.; Gibson, G.R.; Merenstein, D.J.; Pot, B.; Morelli, L.; Canani, R.B.; Flint, H.J.; Salminen, S; Calder, P.C.; Sanders, M.E. The International Scientific Association for Probiotics and Prebiotics consensus statement on the scope and appropriate use of the term probiotic. Rev. Gastroenterol. Hepatol. 2014, 11, 506–514.

  2. the quality of the figure 1 is not good enough
    Response: Thanks for the suggestion. We have added the description and improved the quality of the figure 1.

Plant-based carbohydrate

Starch and

simple sugars

Non-starch

polysaccharides

Hemicellulose

and pectin

Cellulose

Lignin

The starch* and simple sugars contained in plant could be digested by animal digestive tract and provide gross energy. Usually soluble.

Non-starch polysaccharides (NSP) are the parts of plant-base fiber that cannot be digested by animal endogenous digestive enzymes. Soluble NSP can enhance the growth of intestinal microorganisms. Not necessarily soluble. Dietary fiber.

Hemicellulose including xylan, glucan, and mannans, etc., contains lower sugar units than cellulose and is easily degraded by microorganisms and various hemicellulases.

Hemicellulose can combine with pectin and stabilize the strength of cell wall. Not necessarily soluble. Neutral detergent fiber.

Cellulose is ubiquitous in plant cell walls and is primarily responsible for support. Can be degraded by cellulose. Not easily degraded by microorganisms. Insoluble. Acid detergent fiber.

Lignin is present in the cell wall of plants and is mainly responsible for support. It is very strong and can only be degraded by a few enzymes such as laccase. Difficult to be degraded by microorganisms (except fungi). Insoluble.

  • Figure 1. The classification and characteristics of carbohydrates in plants. *Resistant starch is beyond the scope of this figure. The figure information adapted from Prasad and Bondt [12] and International Scientific Association for Probiotics and Prebiotics (ISAPP) organization [19].

  1. Was the figure 1 developed by the authors? If so, include the references where the information has been obtained from. If the figure has been taken from another author, please include a sentence specifying that you have the rights to use it.

Response: Thanks for the suggestion. We have added the description and improved the quality of the figure 1 as above.

  1. such as which ones?
    Response: Thanks for the suggestion. We have corrected and added the description as “Therefore, soluble fiber (3% pectin) may also promote excessive growth of microorganisms (total count of bacteria and coli) and increase chyme viscosity [14]”.

  2. All of the phytochemicals and anti-nutritional compounds listed here are present in high-fiber by-products?
    Response: Thanks for the suggestion. We have corrected and added the description as:

    Partial plant phytochemicals and anti-nutrition factors in high-fiber by products
    In addition to fiber, plants also contain a variety of secondary metabolites, including antioxidant peptides, phenols, flavonoids, phytic acid, trypsin inhibitors, and lectins [22,23]. These plant-derived secondary metabolites are rich in antioxidant, anti-inflammatory, and antibacterial activity, but may also reduce the animal’s absorption efficiency to nutrients such as minerals or amino acids [22]. Phytochemicals have been shown to positive antioxidant benefits toward animals in terms of favored growth performance, production quality, and enhanced endogenous antioxidant system, possibly by directly affecting specific molecular targets or/and indirectly as stabilized conjugates affecting the metabolic pathways [11,13,23]. Accordingly, dissecting the antioxidant effects and the underlying mechanism of dietary phytochemicals is an important area. Therefore much of attention is being focused on a new wave of nutrigenomics [6,10,23]. Several studies dedicated to understand and formulate mechanistic pathways by which these naturally-derived substances could alter the fate of cells, particularly the antioxidant properties of phytochemicals have been implicated as stress-alleviation agents [11,23,24]. Besides, phytic acid, trypsin inhibitors, and lectins, there are also anti-nutritional factors commonly found in plants [24,25] and cannot be ignored while using high-fiber agricultural by-products as animal feed or feed additive.

  3. Could you explain in more detail the mechanism by which ~
    Response: Thanks for the suggestion. We have corrected and added the description and references as “In addition to fiber, plants also contain a variety of secondary metabolites, including antioxidant peptides, phenols, flavonoids, phytic acid, trypsin inhibitors, and lectins [22,23]. These plant-derived secondary metabolites are rich in antioxidant, anti-inflammatory, and antibacterial activity, but may also reduce the animal’s absorption efficiency to nutrients such as minerals or amino acids [22]. Phytochemicals have been shown to positive antioxidant benefits toward animals in terms of favored growth performance, production quality, and enhanced endogenous antioxidant system, possibly by directly affecting specific molecular targets or/and indirectly as stabilized conjugates affecting the metabolic pathways [11,13,23]. Accordingly, dissecting the antioxidant effects and the underlying mechanism of dietary phytochemicals is an important area. Therefore much of attention is being focused on a new wave of nutrigenomics [6,10,23]. Several studies dedicated to understand and formulate mechanistic pathways by which these naturally-derived substances could alter the fate of cells, particularly the antioxidant properties of phytochemicals have been implicated as stress-alleviation agents [11,23,24]. Besides, phytic acid, trypsin inhibitors, and lectins, there are also anti-nutritional factors commonly found in plants [24,25] and cannot be ignored while using high-fiber agricultural by-products as animal feed or feed additive.”

  4. of which?
    Response: Thanks for the suggestion. We have added the description as ” On the other hand, some phenolic compounds can also inhibit the production of pathogenic bacteria such as grape extracts was effective at inhibiting antibiotic-resistant Staphylococcus aureus and E coli, including methicillin-resistant  aureus, with  minimum inhibitory concentration (MIC) ranging from 0.3 to 3.0 mg/mL, by chelating minerals necessary for the microorganism survival, or perforating the microbial cell membranes [28]”

  5. Which ones ? and what is their mechanism of action?
    Response: Thanks for the suggestion. We have added the description as ” However, some plants also contain potentially toxic phenolic compounds, such as Pyrogallol, a simple phenolic found in green tea, was shown to cause hepatic damage when administered at 100 mg/kg in rats. Serum enzymes including aspartate aminotransferase and alanine amino-transferase (ALT) as well as malondialdehyde (MDA) were increased, suggesting that free radical formation and pro-oxidant toxicity played a role. Therefore excessive supplement may cause animal poisoning, shock, and reduced nutrient absorption in animals [29].

  6. how? and what physiological condition?
    Response: Thanks for the suggestion. We have added the description as ” Phenols or flavonoids, the plant secondary metabolites (functional phytochemicals) which the ability to induce the expression of antioxidant/phase II enzymes, it appears that the major role in acting as modifiers of signal transduction pathways to elicit its cytoprotective responses through suppressing stress-induced protein activation and enhancing Kelch-like ECH associating protein 1 (Keap1), a cytoskeleton binding protein, dissociation from nuclear factor (erythroid-derived 2)-like 2 (Nrf2) in response to stressors. Therefore, suppression of abnormally amplified oxidation signaling and restoration of improperly working as well as activation of antioxidant machinery could provide important strategies for prevention of oxidative stress and augmentation of antioxidant defense in animals [3,15].

  7. [36]leading
    Response: Thanks for the suggestion. We have corrected description as” [36] leading”.

  8. Italics

Response: Thanks for the suggestion. We have corrected description as” epigallocatechin gallate and epigallocatechin”.

  1. Specify
    Response: Thanks for the suggestion. We have corrected and added the description as “Therefore, for the utilization of lectins, their physicochemical properties (including hemagglutination activities inflammatory, antibacterial, and antifungal activity) as well as dosage must be considered and evaluated”.

  2. Please avoid using the term "Flora" or "microflora" throughout the document, as it is incorrect.
    Response: Thanks for the suggestion. We have corrected the description as “microbiota” throughout the document.

  3. microflora
    Response: Thanks for the suggestion. We have corrected the description as “microbiota” throughout the document.

  4. supplements
    Response: Thanks for the suggestion. We have corrected the description as “replacement or addition”.

  5. microbial flora
    Response: Thanks for the suggestion. We have corrected the description as “microbiota”.

  6. Limit the use of italics to scientific names.
    Response: Thanks for the suggestion. We have corrected the description in the Table 1.

  7. add (log CFU)
    Response: Thanks for the suggestion. We have added the description as “(log CFU)” in the Table 1.

  8. Laying hens and other poultry would still need to be addressed.
    Response: Thanks for the suggestion. We have corrected and added the description “Effects of fiber on broilers”.

  9. (1-14-day-old) would
    Response: Thanks for the suggestion. We have corrected the description “Kermanshahi et al [14] indicated that the 3 % cellulose addition in the diet of broilers (Ross 308) had similar growth performance, intestinal morphology, microbe composition, and serum characteristics in comparison to the control group during the experimental period (0 to 14 day).”

  10. Structure
    Response: Thanks for the suggestion. We have corrected the description as “composition”.

  11. Lee et al have discussed in detail the inflammation, antioxidant mechanism and phytogenics, and fungus type probiotics on animal immune regulation and antioxidant capacity [69,70,10].
    Response: Thanks for the suggestion. We have corrected the description as “Previous studies have discussed in detail the inflammation and antioxidant mechanism of phytochemicals on animal immune regulation and antioxidant capacity [69,70,10]. Briefly, when animals are subjected to environmental stress, such as heat stress or pathogens infection, stimulation causes an inflammatory response and increases oxidative stress in animals [71]. Stimulated by pathogenic bacteria, the animal initiates an immune response leading to a cytokine storm [71]. However, excessive inflammation can reduce animal performance and even lead to death [10]. On the other hand, when the oxidative pressure is too high, animals are not able to eliminate the damage caused by free radicals to cells or organs [69]. Among them, the animal's antioxidant system is mainly regulated by the liver, so the antioxidant capacity is also related to liver performance [3]. Applying phytochemicals or botanical compounds to the feed could promote intestinal health, reduce the inflammatory response and enhance the antioxidant capacity of the animal [10]. The mechanisms are majorly the nuclear factor (erythroid-derived 2)-like 2 (Nrf2) and nuclear factor kappa B (NF-κB) for they are respectively the key transcription factors involved in oxidative stress and inflammation for elucidating the underlying signal transduction pathways. Therefore, phytochemicals can regulate these transcription factors leading to the improvement of oxidative status, heme oxygenase-1 (HO-1) gene is found crucial for Nrf2-mediated NF-κB inhibition. Hence, proper fibers as phytochemicals (likely 0.5-1% mulberry leaves addition in laying hens) with such modulatory effects worth be used to explore the possible crosstalk in oxidative stress and immunomodulation in animals [69-71].”

  12. what type of fiber? in what quantity?
    Response: Thanks for the suggestion. We have corrected the description as ” Hence, proper fibers as phytochemicals (likely 0.5-1% mulberry leaves addition in laying hens) with such modulatory effects worth be used to explore the possible crosstalk in oxidative stress and immunomodulation in animals [69-71].”.

  13. what type of fiber? in what quantity?
    Response: Thanks for the suggestion. We have corrected the description as ” Giving a high-fiber (totally dietary fiber about 28.2%), low-energy diets (including mainly 24.4% soybean hulls) to sows in group cultures can improve their feeding time and health, and reduce aggressive and stereotyped behavior [73,74].”.

  14. Giving higher amount of fiber can also improve the welfare and improve the production performance of sows [77] and are, therefore, beneficial to the health of pregnant sows. However, the high fiber content in the feed does not benefit the pellets. Because of the fluffy characteristic of fiber, the stability of pellets is destroyed. Fiber addition
    Response: Thanks for the suggestion. Concerning the sentence is not related to the “Fibers satiety in animals” section to other reviewer, we have deleted and corrected the description as “Providing higher amount of fiber (7.5% crude fiber consist of majorly 20% Alfalfa meal and 52% corn in lactation diet) can also improve the welfare and improve the production performance of sows [77] and are, therefore, beneficial to the health of pregnant sows.”

  15. Limit the use of italics to scientific names.
    Thanks for the suggestion. We have corrected the description in the Table 2.

  16. The information in this paragraph is outside the scope of this review, as it does not refer to agricultural by-products.
    Response: Thanks for the valuable suggestion. We have deleted the paragraph description.

  17. what type of fiber? in what quantity?
    Response: Thanks for the suggestion. We have added the description as “addition of prebiotics (autoclaved drinking water supplements with 1.0% oligofructose-enriched inulin (w/v))”.

  18. Specify how much is excessive
    Response: Thanks for the suggestion. We have corrected the description “However, excessive fiber addition (more than 40% peach palm (PP) meal replacement for maize in goats, the NDF corrected for ash and protein (NDFap) is 40.1% and acid detergent fiber (ADF) is 20.2 % in 40% DM level of PP meal substitution group) may still cause a decrease in palatability, and the effect of digestibility is also related to the source of fiber [91].”

  19. The conclusions???
    Response: Thanks for the suggestion. We have corrected the description as “Conclusions”.

  20. Unnecessary
    Response: Thanks for the suggestion. We have deleted the description.

Overall, still very much thanks for the reviewer’s response and comments.

Round 2

Reviewer 2 Report

I accept the paper in this form. All remarks included in previous review have been taking into consideration by paper authors and implemented in the manuscript.

Author Response

Responses to Reviewer II 's comments:

I accept the paper in this form. All remarks included in previous review have been taking into consideration by paper authors and implemented in the manuscript.

Response: We really appreciate the comments, and thank you for providing the useful suggestion.

Reviewer 4 Report

Dear authors
Thanks for you answer. I only insist that the abbreviations section is not required.

Author Response

Responses to Reviewer IV 's comments:

Dear authors,

Thanks for you answer. I only insist that the abbreviations section is not required.
Response: We really appreciate the comments and the suggestion. We have deleted the abbreviations section.
